# Planning for Proactive Assistance in Environments with Partial Observability

**Anagha Kulkarni, Siddharth Srivastava, Subbarao Kambhampati**

Department of Computer Science, Arizona State University
{anaghak, siddharths, rao}@asu.edu

## Abstract

This paper addresses the problem of synthesizing the behavior of an AI agent that provides proactive task assistance to a human in settings like factory floors where they may coexist in a common environment. Unlike in the case of requested assistance, the human may not be expecting proactive assistance and hence it is crucial for the agent to ensure that the human is aware of how the assistance affects her task. This becomes harder when there is a possibility that the human may neither have full knowledge of the AI agent's capabilities nor have full observability of its activities. Therefore, our *proactive assistant* is guided by the following three principles: **(1)** its activity decreases the human's cost towards her goal; **(2)** the human is able to recognize the potential reduction in her cost; **(3)** its activity optimizes the human's overall cost (time/resources) of achieving her goal. Through empirical evaluation and user studies, we demonstrate the usefulness of our approach.

## 1  Introduction

While assisting the humans may be tricky for an AI agent even when the humans explicitly request for assistance, it is even more challenging for the agent to provide the assistance when it has to do it proactively. Not only does it have to reason over the human's goals to synthesize an assistive behavior that reduces the human's costs, but it also has to make sure that the assistance it provides can be recognized by the human, who may not be expecting it. Like the proverbial justice, *proactive assistance should not only be provided, but should be seen to be provided*. This further becomes challenging in environments where the human may have partial observability of the AI agent's activities. The agent thus needs to synthesize communicative behaviors – be they purely epistemic (speech acts) or ontic actions with epistemic effects – which allow the human to recognize the assistance. This requires it to control the human's observability by reasoning over her belief states.

This paper specifically looks at the problem of providing proactive assistance to a human in an environment where the AI agent and the human coexist, and have partial observability of each other's activities. There are several real-world workspaces like factory floors, warehouses, restau-

rants, nursing homes for elderly, disaster response areas, etc., where this problem of providing proactive task assistance to the involved humans is important. Our formulation considers a scenario where the AI agent is aware of the tasks being allocated to the human by the ecosystem and may also know the rules and protocols of the ecosystem. We assume that the agent has access to an input that captures the human's planning process for her goals. For instance, prior works that study the problem of action model acquisition (Zhuo and Yang 2014; Zhuo and Kambhampati 2013) can be used to derive the human's planning process. This allows it to synthesize assistive plans and to reason over the impact of those plan on the human's goals and plans. This leads us to the first principle: **(1)** *A proactive assistant's behavior should only decrease the human's optimal cost towards her goal*.

Further, since the agent is providing proactive assistance, it is essential for it to ensure that the human recognizes the assistance and modifies her original plan towards her goal. Additionally, our formulation accommodates environments where both the human and the AI agent may not have full visibility of each other's activities. For instance, the human may not know what activities were performed by a robot in another room. Therefore, it should be able to take into account the human's perception limitations as well as the possible ways of communicating necessary information to her. This leads us to the second principle: **(2)** *A proactive assistant should make the human aware of the potential reduction in her cost as a result of its assistance*.

Lastly, it is important to capture the cost of the assistance to the human. For instance, if the human has to wait for a really long time for the agent to provide assistance, then the human may instead prefer to work by herself. Since, the human is actively involved in the overall plan, it is not only necessary to reduce the human's cost to her goal, but also to reduce her overall effort resulting from processing the agent's behavior. Therefore, **(3)** *A proactive assistant should optimize for the overall cost incurred by the human in terms of the time taken (or resources needed) to participate in the overall plan*. Together these principles guide our proactive assistant. In the following sections, we propose a Monte Carlo Tree Search (MCTS) based solution that modulates the human's belief by either communicating necessary information or limiting irrelevant information (i.e. by controlling

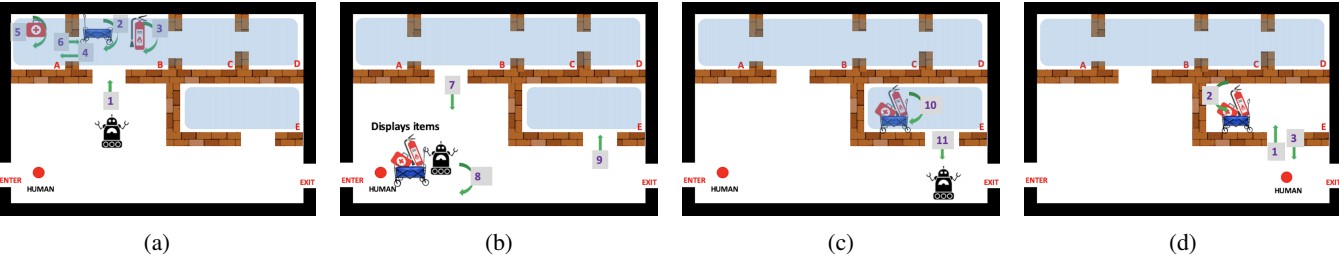

Figure 1: Illustration of an assistive joint plan in urban search and rescue domain. (a) The robot collects items required for a side goal (fire extinguisher) and human's goal (medkit) in a wagon, (b) makes the human aware of the items it is carrying by showing them, (c) leaves the wagon in Room E. (d) The human collects the medkit from room E to accomplish her goal.

human's observability) to communicate the potential cost reduction to the human. We perform an empirical evaluation and a user study to assess the utility of our approach.

**Running Example** Let's consider a concrete example in an urban search and rescue domain. Here a human commander and a robot are operating on a floor as shown in the Figure 1a. The human's task is to find a medkit on this floor. She has access to the floor map but does not know where the medkit is. Hence, her cost for accomplishing the task is very high (she has to search each room on the floor). The robot is aware of the task allocated to the human. It is also working on a non-urgent side goal of dropping a fire extinguisher to room E (as shown in Figure 1c). The robot who is already operating on that floor has more information about the locations of the items and is capable of assisting the human. However, since the assistance is being provided proactively, it is important for the robot to ensure that the human can recognize how the assistance optimizes her task. The robot assists as detailed in Figure 1a to 1c, where it first collects the medkit in its wagon, and then it performs an action to display the contents of the wagon to the human and then transports it to room E. This allows the human to form a belief that the medkit is in room E. She can now optimize her goal, as shown in Figure 1d.

## 2 Related Work

The problem of synthesizing a proactive assistant is directly connected to the prior literature on modeling assistive agents. Starting from the seminal work on SharedPlan theory (Grosz and Kraus 1996) which discussed the notion of "intentions-to" and "intentions-that" constructs to more recent work on the development of a theory of assistance, there has been a lot of research in this general direction. In the SharedPlan theory (Grosz and Kraus 1996), the "intentions-that" constructs refer to the acts that are performed by an agent as a responsibility towards other agents. In our framework, the AI agent's assistive actions towards the human fall under this category. In some of the more recent works (Fern et al. 2014; Oh et al. 2010) the emphasis has been on learning the human's goal first by performing information gathering actions and following those with assistive actions towards the goal. We build on these works by assuming the human's goal is known beforehand and emphasize on how

the agent can ensure the communication of its proactive assistance. Further, work by (Yorke-Smith et al. 2012) delineates desired properties like pertinence, competency, etc., of a proactive assistant and proposes an operational framework. Our concrete guidelines can be traced back to these general guidelines. Other works (Kamar, Gal, and Grosz 2009; Unhelkar and Shah 2016) have studied in a collaborative setting with predefined roles for agents, whether an agent should help others or not. In our case, the agent stops itself from assisting proactively only when the assistance puts the human in a worse-off situation.

More recently, the idea of planning for stigmergic collaboration in human-agent cohabitation scenarios (Chakraborti et al. 2015; Buckingham and Scheutz 2017) has been explored. However, in these works, the agent does not reason over human's awareness of the assistance, in addition human's partial observability of the agent's actions is not considered. Further, unlike some decision support systems (Sengupta et al. 2017) that tend to be proactively assistive at planning time by providing plan suggestions, our system provides proactive task assistance at execution time. Assistance at execution time has additional challenges of managing human's observability. Besides managing human's observability, sometimes it might be necessary to manage her attention. For instance, despite the agent's attempt at making the human aware of the assistance, she may voluntarily or involuntarily be inattentive, thereby invalidating agent's efforts. This aspect of attention management (Horvitz, Jacobs, and Hovel 2013) has been studied in the literature. However, we do not consider the problem of human's attention management here. Lastly, we borrow the notion of controlled observability from (Kulkarni, Srivastava, and Kambhampati 2019). In that work, they use legible behaviors (Dragan and Srinivasa 2013; MacNally et al. 2018; Miura and Zilberstein 2020) to communicate in cooperative scenarios and obfuscatory behaviors (Keren, Gal, and Karpas 2016; Masters and Sardina 2017) to hide in adversarial scenarios. However, we show in the upcoming sections that both legibility and obfuscation can be used to communicate assistance.

## 3 Problem Framework

We consider two actors $\mathbf{R}$ (say, a robot) and $\mathbf{H}$ (say, a human). The objective of $\mathbf{R}$ is to proactively provide task level assistance to $\mathbf{H}$ at execution time. As mentioned before, by

using **H**'s decision algorithm as an input to our system, we can simulate **H**'s plans.

**Planning**   We use the notations of planning problems (Geffner and Bonet 2013) to define our framework. A planning problem can be defined as a tuple $\mathcal{P} = \langle \mathcal{F}, \mathcal{A}, \mathcal{I}, \mathcal{G}, \mathcal{C} \rangle$, where $\mathcal{F}$, is a set of fluents, $\mathcal{A}$, is a set of actions, and $c$ is the cost for each action. A state $s$ of the world is an instantiation of all fluents in $\mathcal{F}$. Let $\mathcal{S}$ be the set of states. $\mathcal{I} \in \mathcal{S}$ is the initial state, that is all the fluents are instantiated. $\mathcal{G}$ is the goal where a subset of fluents in $\mathcal{F}$ are instantiated. Each action $a \in \mathcal{A}$ is a tuple of the form $\langle pre(a), add(a), del(a) \rangle$ where $pre(a) \subseteq \mathcal{F}$ is a set of preconditions, $add(a) \subseteq \mathcal{F}$ is a set of add effects and $del(a) \subseteq \mathcal{F}$ is a set of delete effects of action $a$. The transition function $\Gamma(\cdot)$ is given by $\Gamma(s, a) \models \perp$ if $s \not\models pre(a)$; else $\Gamma(s, a) \models s \cup add(a) \setminus del(a)$. The solution to $\mathcal{P}$ is a *plan* or a sequence of actions $\pi = \langle a_1, a_2, \ldots, a_n \rangle$, such that, $\Gamma(\mathcal{I}, \pi) \models \mathcal{G}$, i.e., starting from the initial state and sequentially executing the actions results in the robot achieving the goal. The cost of the plan, $\mathcal{C}(\pi)$, is a sum of the cost of all the actions in it, $\mathcal{C}(\pi) = \sum_{a_i \in \pi} \mathcal{C}(a_i)$.

## 3.1  MA-COPP

In our setting, **R** is aware of both its own model and **H**'s model. Whereas **H** is only aware of its own model. Both the agents have full observability of their own activities. However, they both have partial observability of certain actions performed by the other agent. **R** is also aware of **H**'s perception limitations of its actions[1] and is capable of choosing among multiple actions to modulate **H**'s observability of its actions. We call this framework as *multi-agent controlled observability planning problem* (or MA-COPP).

**Definition 1.** *A **multi-agent controlled observability planning problem** is a tuple,* MA-COPP $= \langle \mathcal{P}_{\mathbf{H}}, \mathcal{M}_{\mathbf{R}}, \Omega_{\mathbf{H}}, \mathcal{O}_{\mathbf{H}} \rangle$,

- $\mathcal{P}_{\mathbf{H}} = \langle \mathcal{F}, \mathcal{A}_{\mathbf{H}}, \mathcal{B}_0, \mathcal{G}_{\mathbf{H}}, \mathcal{C}_{\mathbf{H}} \rangle$ *is **H**'s planning problem. $\mathcal{B}_0$ is its initial belief, which is a set of states inclusive of actual initial state $\mathcal{I}$.*

- $\mathcal{M}_{\mathbf{R}} = \langle \mathcal{F}, \mathcal{A}_{\mathbf{R}}, \mathcal{I}, \mathcal{C}_{\mathbf{R}} \rangle$ *is **R**'s action model. **R** has full observability of its actions and states.*

- $\Omega_{\mathbf{H}}$ *is the set of observation symbols received by **H**, when it acts or when **R** acts.*

- $\mathcal{O}_{\mathbf{H}} : \mathcal{A}_{\mathbf{R}} \cup \mathcal{A}_{\mathbf{H}} \times \mathcal{S} \to \Omega_{\mathbf{H}}$ *is **H**'s sensor model. Further, $\exists a, a' \in \mathcal{A}_{\mathbf{R}}, \ s, s' \in \mathcal{S}, a \neq a' \wedge s \neq s' : \mathcal{O}_{\mathbf{H}}(a, s) = \mathcal{O}_{\mathbf{H}}(a', s')$, i.e., $\mathcal{O}_{\mathbf{H}}$ gives coarse-grained observations for at least some actions of **R**, making some of **R**'s actions seem indistinguishable.*

From the definition of MA-COPP, we can see that although both the agents have independent action models, action costs, they share the same state space. Moreover, **H**'s initial belief consists of **R**'s initial state. To select a specific behavior that modulates **H**'s information in the environment, **R** requires access to **H**'s prior knowledge, its perception limitations as well as its task. In the running example, this involves modeling the fact that the human does not

---

[1] We consider a single-interaction assistive setting and therefore do not model **R**'s partial observability of **H**'s actions.

know the location of the items, as well as that she cannot see the actions performed in other rooms, and that her goal is to find a medkit on that floor.

In MA-COPP, **R** executes an assistive behavior from the initial state followed by **H**'s execution from the updated belief towards its goal. For instance, in the running example, the robot displays the wagon and leaves it in room E followed by the human commander's execution of her plan. Due to **H**'s partial observability of **R**'s actions, it operates in a belief space. For instance, the human didn't know the contents of the wagon before they are displayed. In solving MA-COPP, the challenge lies in choosing the right amount of information to reveal to **H**. **R** can select actions that reveal the missing information to **H**. It can also select actions that hide away unnecessary complexities from **H**. Therefore, in solving MA-COPP, **R** has to carefully choose what information to reveal versus what to hide from **H**.

**R's Modeling of H's Belief Update.**   After an action $a \in \mathcal{A}_{\mathbf{R}} \cup \mathcal{A}_{\mathbf{H}}$ changes the current state of world resulting in a new state $s \in \mathcal{S}$, **R** simulates **H**'s belief update by using the observation $\mathcal{O}_{\mathbf{H}}(a, s)$ emitted by **H**'s sensor model. The definition of **H**'s sensor model also allows for actions with null observations. That is, a dummy observation, $\omega^{\emptyset} \in \Omega_{\mathbf{H}}$, that makes all **R**'s $\langle a, s \rangle$ pairs seem indistinguishable, where $a \in \mathcal{A}_{\mathbf{R}}, s \in \mathcal{S}$. At any time step, $t \in \{1, \ldots, \mathcal{T}\}$, $\mathcal{B}_t$ represents **H**'s belief. Here a belief essentially is a set of states, and $\mathcal{T}$ is the last time step of the joint execution. If $|\mathcal{B}_t| = 1$, then **H** has full observability of the state at time step $t$. The belief update is defined as follows: **(1)** at time step $t = 0$, $\mathcal{B}_0 = \{s \mid \exists s \in \mathcal{S}, s = \mathcal{I}\}$, **(2)** at time step $t \in \{1, \ldots, \mathcal{T}\}$, let $\omega_t^{\mathbf{H}} \in \Omega_{\mathbf{H}} \setminus \omega^{\emptyset}$ be the observation received by **H**, then $\mathcal{B}_t = \{s' \mid \exists s \in \mathcal{B}_{t-1}, a \in \mathcal{A}_{\mathbf{R}} \cup \mathcal{A}_{\mathbf{H}} : \Gamma(a, s) \models s' \wedge \mathcal{O}_{\mathbf{H}}(a, s') = \omega_t^{\mathbf{H}}\}$. If $\omega_t^{\mathbf{H}} = \omega^{\emptyset}$ then $\mathcal{B}_t = \mathcal{B}_{t-1}$. This is because practically the null observation does not reveal any new information in **H**'s belief update. **H** starts its execution from an intermediate belief state. Let $t = k$ be an intermediate time step and $\mathcal{B}_k$ be **H**'s starting belief for its execution. **R** maintains **H**'s belief state from initial belief until $t = k$.

**Formal Guidelines for a Proactive Assistant**   An implicit objective of **R** is to ensure that **H**'s cost of achieving its goal is less than that of achieving its goal by itself. We can formalize this intuition about loss/gain in terms of cost experienced by **H** when it participates in a joint execution by using the notion of cost differential. Given a joint plan, $\pi_{\text{MA-COPP}}$, that solves a planning problem for the goal, $\mathcal{G}_{\mathbf{H}}$, let $\mathcal{C}_{\mathbf{H}}^{\Delta}(\pi_{\text{MA-COPP}})$ represent the cost differential between the cost incurred by **H** when $\pi_{\text{MA-COPP}}$ is executed, versus the minimum cost it incurs when it achieves the goal by itself, i.e., $\mathcal{C}_{\mathbf{H}}^{\Delta}(\pi_{\text{MA-COPP}}) = \mathcal{C}_{\mathbf{H}}(\pi_{\text{MA-COPP}}) - \mathcal{C}_{\mathbf{H}}(\pi_{\mathbf{H}}^*)$.

For **H** to participate in an assistive joint plan with **R**, it only makes sense if and only if the assistance provides a reduction in her total cost. Otherwise, **H** may be better off executing its own plan to its goal. Therefore, for **R** to be an assistive agent, the first constraint is to ensure that it only produces a joint plan where the assistance decreases **H**'s minimum cost (given by **H**' decision algorithm). That is,

for a joint plan $\pi_{\text{MA-COPP}}$, $\mathcal{C}_{\mathbf{H}}^{\Delta}(\pi_{\text{MA-COPP}}) < 0$. In addition, $\mathbf{R}$ should keep track of the belief updates that $\mathbf{H}$ may go through before the start of its execution phase. Given that, $\mathbf{R}$ is aware of $\mathbf{H}$'s sensor model, by simulating the belief it can choose its actions to either limit or increase the amount of information being shared with $\mathbf{H}$. $\mathbf{R}$ can achieve this in multiple ways: (1) by either making certain part of the current state legible (collapsing the states in $\mathbf{H}$'s belief) to reveal particular information to $\mathbf{H}$, or (2) by obfuscating the current state completely thereby keeping some unnecessary complexities hidden from $\mathbf{H}$'s belief. This belief modulation allows $\mathbf{H}$ to participate in the joint plan. As without any awareness about the assistance, $\mathbf{H}$ may tend to follow her original plan. Thus by controlling $\mathbf{H}$'s observability, $\mathbf{R}$ can not only assist $\mathbf{H}$ but also guide it towards a cheaper plan to $\mathcal{G}_{\mathbf{H}}$. As a result of this belief modulation, $\mathbf{H}$'s planning problem gets modified to $\mathcal{P}_{\mathbf{H}}^k = \langle \mathcal{F}, \mathcal{A}_{\mathbf{H}}, \mathcal{B}_k, \mathcal{G}_{\mathbf{H}}, \mathcal{C}_{\mathbf{H}} \rangle$, where $k$ represents the number of actions executed by $\mathbf{R}$. Let $\pi_{\mathcal{P}_{\mathbf{H}}^k}$ be a minimum cost plan (as per $\mathbf{H}$' decision algorithm) for this modified problem, then $\mathcal{C}_{\mathbf{H}}(\pi_{\mathcal{P}_{\mathbf{H}}^k}) = \mathcal{C}_{\mathbf{H}}^{\Delta}(\pi_{\text{MA-COPP}})$. Therefore, the second constraint for $\mathbf{R}$ is that, $\mathbf{H}$ should be aware of the reduction in its cost in the modified planning problem.

Finally, the overall effort needed from $\mathbf{H}$'s end to participate in $\pi_{\text{MA-COPP}}$ should be minimized while accounting for both the prior constraints. This is important because, even though $\mathbf{H}$ only starts executing after $\mathbf{R}$, $\mathbf{H}$'s active involvement in the joint plan itself starts from the beginning of the plan. This involves the additional overhead experienced by $\mathbf{H}$ in updating its belief as a result of $\mathbf{R}$'s actions. This penalty incurred by $\mathbf{H}$ can be formulated in different ways (for e.g., the cost associated with belief update during $\mathbf{R}$'s execution, etc). We approximate this penalty as the overall length (time steps) of $\mathbf{R}$'s part of the joint execution[2] in addition to $\mathbf{H}$'s execution cost. Let $\mathcal{L}$ be the maximum cost that $\mathbf{H}$ is willing to accommodate in the first part of the joint execution, i.e., $k < \mathcal{L}$. Therefore, a proactive assistant optimizes:

$$\min \quad \alpha\, k + (1 - \alpha)\, \mathcal{C}_{\mathbf{H}}^{\Delta}(\pi_{\text{MA-COPP}}) \tag{1}$$

$$\text{subject to} \quad \mathcal{C}_{\mathbf{H}}^{\Delta}(\pi_{\text{MA-COPP}}) < 0 \tag{2}$$

$$\mathcal{C}_{\mathbf{H}}(\pi_{\mathcal{P}_{\mathbf{H}}^k}) = \mathcal{C}_{\mathbf{H}}^{\Delta}(\pi_{\text{MA-COPP}}) \tag{3}$$

$$k < \mathcal{L} \tag{4}$$

where $\alpha$ is a parameter. By setting $\alpha$ appropriately, we can choose joint plans that $\mathbf{H}$ may prefer in terms effort required.

## 4 Solution Methodology

Although, the overall objective here is to find a joint plan that satisfies equation 1, we can only synthesize behavior of the autonomous agent, $\mathbf{R}$. We assume that $\mathbf{H}$ is an independent agent capable of planning towards its own goal. Given that $\mathbf{H}$ has partial observability of some of the actions performed by $\mathbf{R}$ and operates in a belief space, we assume that $\mathbf{H}$ is capable of computing a conformant plan (Hoffmann and Brafman 2006b; Palacios and Geffner 2009) from

---

[2]Instead of using the entire length of $\mathbf{R}$'s part of the joint plan, we can choose to use only the length of observable time steps, i.e. we can choose to ignore the time steps with null observations.

the belief at the beginning of its execution phase. A conformant plan solves the task by accounting for the relevant uncertainties and does not rely upon being able to get further information from $\mathbf{R}$.

Therefore, a solution to equation 1 involves finding a plan for $\mathbf{R}$ from the initial state to a desirable belief state, $\mathcal{B}_k$ considering the best response of $\mathbf{H}$ at time step $k$. Practically, this is a nested search process, where in the outer search loop, the algorithm searches for a desirable belief state by performing a sequence of actions consistent with $\mathbf{R}$'s action model. While in the inner search loop, the algorithm searches for satisfaction of the goal by performing a sequence of actions consistent with $\mathbf{H}$'s action model. However, since $\mathbf{H}$ operates in a belief space, for each node we need to maintain $\mathbf{H}$'s belief consistent with that node. And this nested search is essentially a search over a belief space that not only achieves the goal, but also reaches an intermediate partially legible or obfuscatory belief. Since it is not known beforehand, what a desirable belief state would look like for a given problem, it is not that straightforward to design a goal-directed heuristic function to expand the search space. Instead, we use Monte Carlo tree search (MCTS) as a possible way of quickly sampling states and building a utility based tree by performing simulations using a conformant planner for the inner search loop. Once we have access to such a tree, we can then perform search on it by expanding only the high utility search nodes in the tree.

In our approach, we only synthesize for a single agent in a serialized manner. Therefore, there is no need to wait for the moves of the second actor and we can use a single-player version of MCTS (Schadd et al. 2008). By running numerous quick simulations on the solution space, we can build a sufficiently good utility tree starting from initial state of $\mathbf{R}$. The single player MCTS approach for constructing the utility tree is outlined in Algorithm 1. For $n$ iterations, the selection of nodes to be expanded in the tree is done using UCT (upper confidence bound 1 applied to trees) given by $\frac{\text{node utility}}{\text{node visits}+\epsilon} + C * \sqrt{\frac{\ln \text{elapsed iterations}}{\text{node visits}}}$ (Kocsis and Szepesvári 2006). The depth of the tree is expanded until $\mathbf{R}$'s budget $\mathcal{L}$ (from Equation 4) runs out. For each of the expanded nodes, we simulate using a conformant plan generated from node's belief to solve $\mathcal{G}_{\mathbf{H}}$. The satisfaction of the goal and the length of the plan, determines the overall reward to be backpropagated.

The utility tree thus constructed is then used to compute the actual joint plan. In this utility tree, we can consider $n$ best children for each node (i.e. nodes with higher utility and/or higher number of visits). This helps in reducing the solution space with paths that have now been sampled to ensure the satisfaction of all the constraints listed out in equations 2 through 4. On this reduced search space, we can now perform a simple search to find a node that minimizes the equation 1 as well as satisfies the goal. The path to the best such node is then the part of the joint plan that is executed by $\mathbf{R}$. This secondary search on the utility tree is only to ensure that the solution minimizes the equation 1. Additionally, $n$ can be increased to ensure completeness. Depending on the number of iterations $m$ of MCTS, the value of $n$ can

**Algorithm 1** Generation of utility tree

1: **Input:** MA-COPP, $\mathcal{C}_{\mathbf{H}}(\pi_{\mathbf{H}}^*)$, $\mathcal{L}$, $m$ (number of iterations), $\beta$ (reward constant),
2: $\phi$ (cost constant)
3: **Output:** $tree$ (utility tree)
4: $tree \leftarrow node(\mathcal{I}, \mathcal{B}_0, utility = 0)$
5: **for** $m$ iterations **do**
6:     /* select a leaf node using UCT to evaluate nodes */
7:     $node, t_{node} = \text{select}(tree)$
8:     /* expand a child node */
9:     $child, t_{child} = \text{expand}(node, t_{node})$
10:     **if** $t_{child} < \mathcal{L}$ **then**
11:         /* create conformant planning problem */
12:         $\pi_{\mathbf{H}} = \text{planner}(child.\mathcal{B}_t)$
13:         /* simulate using conformant plan */
14:         **if** $\pi_{\mathbf{H}} \neq \emptyset$ & $\mathcal{C}_{\mathbf{H}}(\pi_{\mathbf{H}}) < \mathcal{C}_{\mathbf{H}}(\pi_{\mathbf{H}}^*)$ & $\mathcal{B}_{\mathcal{T}} \models \mathcal{G}_{\mathbf{H}}$ **then**
15:             $reward = \beta$
16:             $cost = \alpha \ t_{child} + (1 - \alpha) \ \mathcal{C}_{\mathbf{H}}(\pi_{\mathbf{H}})$
17:         **else**
18:             $reward = 0$
19:             $cost = \phi$
20:         **end if**
21:         /* Backpropagate reward and cost */
22:         $\text{backpropagate}(child, reward - cost * \epsilon)$
23:     **end if**
24: **end for**

be modulated.

# 5 Evaluation

We conducted a user study to validate the underlying hypothesis of our framework, that the human only recognizes the reduction in her own cost to the goal when the agent takes into account the human's awareness of the assistance. For the user study, we use urban search and rescue (USAR) domain presented in the running example. We also perform an empirical evaluation to analyze the performance of our approach using USAR domain and modified IPC Driverlog domain.

**Domain Setup** For both Driverlog and USAR, we create two versions of the domain: for **R** and **H** respectively. **R**'s version consists of actions that are partially observable as well as non-observable to **H**. Further, for each action, there are two action definitions in the domain: one to capture **R**'s state transition with full observability as well as the other annotated with keyword "belief" to perform the corresponding belief update for **H**. A parser is used to apply either the belief version of the action (for actions without full observability to **H**) or the regular version of the action (for actions with full observability to **H**). In the "belief" version of the actions, to represent uncertainty over some fluents, we use the standard semantics used in conformant planning benchmarks like "unknown", "oneof" clause to mark a fluent as uncertain. For **H**'s version of the domain, some action definitions that depend on uncertain fluents have conditional effects, written using the standard "when" clause consisting of the condition followed by the effect.

**Empirical Evaluation** We use the approach discussed in Section 3 to generate solutions. We use Conformant-FF planner[3] (Hoffmann and Brafman 2006a) to simulate **H**'s plans given a belief state. For both domains, we kept $\mathcal{L} = 15$, i.e., maximum length of **R**'s part of the joint plan. We ran our experiments on 3.5 GHz Intel Core i7 processor with 16 GB RAM. In Figure 4b, we report for each problem **H**'s optimal cost without any assistance from **R**, **H**'s cost from participation in joint plan, percentage decrease in **H**'s cost, length of the joint plan, number of iterations used to construct the utility tree and the time taken to generate the solutions. By setting $\alpha$ parameter, we can see how the joint plans prioritize task load vs processing load. We varied $n$ best children from 1 to 5 during the search but the solutions were not impacted thus indicating that the optimal solutions had been found for those problems for $n = 1$. As shown in the table, a steep percentage decrease is obtained for both domains (specifically for USAR). Additionally, the joint plan itself is not too long even when **R** is assisting. This is because **R** has more information and is capable of guiding **H** in a way that reduces **H**'s cost.

**User Study** We conducted two user studies each with a within-subject design to validate the underlying hypothesis. The participants for the studies were recruited from Amazon Mechanical Turk (Crowston 2012). For each study, we collected 34 submissions. For the first user study after filtering, we had 31 submissions, and for the second we had 27 submissions. Each participant was paid at the rate of $15/hour for 10 minutes.

**Hypothesis 1** *Without legible (revealing information) actions,* **H** *is not aware of the assistance provided by* **R**.

**Hypothesis 2** *Both legible (revealing information) and obfuscating (hiding information) actions allow* **H** *to experience reduction in task load and processing load.*

The format of our studies was as follows: the subjects read through the rules of the USAR domain. Then they were shown two scenarios one after another illustrating the robot's behavior. After seeing each scenario they were asked how they would solve the task (without any interference from the robot). This was also the filter question to make sure they understood the scenario. The submissions which passed the filtering were used to calculate the results. The two scenarios were different in only one action. One scenario satisfied equation 3 (scenario involving proactive assistant), the other did not (baseline scenario). The scenarios that satisfy 3 have been discussed in Figure 4b. The order of the scenarios was flipped for half of the participants to account for sequential bias. In user study 1, Figure 2 with and without the display action (action number 3 in Figure 2b) was shown, while in user study 2, they were shown the illustration in Figure 3 with and without the explanation that the rooms are cleared (action number 3 in Figure 3b). After the filter question, they were asked to answer a survey: **(1)** rate the scenarios

---

[3]Source code for Conformant-FF: https://fai.cs.uni-saarland.de/hoffmann/cff.html

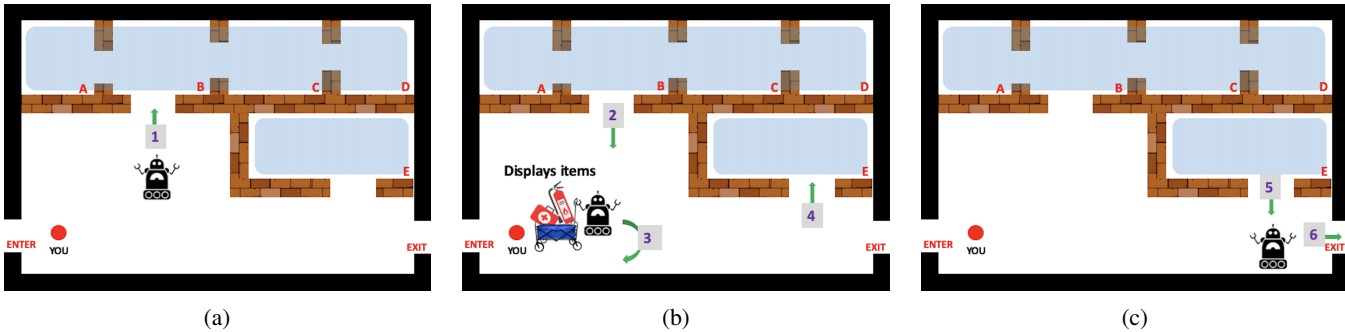

(a)            (b)            (c)

Figure 2: Illustration of assistive plan used in first user study. The goal of the human commander is to find a medkit. She does not know what items are present in each room (indicated by blue regions) (a) The robot goes into room B, (b) comes out with a wagon and shows her the items of the wagon. It then proceeds to room E, (c) comes out without the wagon and exits the floor.

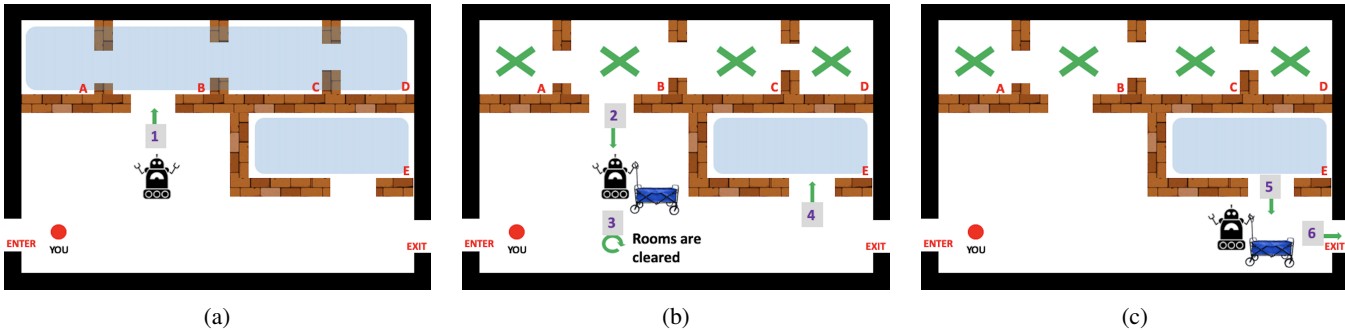

(a)            (b)            (c)

Figure 3: Illustration of assistive plan used in the second user study. Human's goal is to find all the medkits. She does not know what items are present in each room (indicated by blue regions) (a) The robot goes into room B, (b) comes out with a wagon and declares all rooms A, B, C, D are empty. It then proceeds to room E, (c) comes out and exits the floor with the wagon.

in terms of workload **(2)** rate the scenarios in terms of the effort needed to come up with a plan **(3)** what they thought the robot was doing scenario 1 **(4)** same for scenario 2. For questions (1) and (2), they had to rate the scenarios on a 7 point Likert scale "1 – very hard" to "7 – very easy". For questions (3) and (4), they were given the following 4 options - **(a)** working on its task **(b)** assisting them **(c)** working on its task and assisting them **(d)** cannot say.

For the first user study, we used the illustration in Figure 2. This is the same problem as the one explained in the running example, except the participants were not aware of the actions performed by the robot within the rooms. In this scenario, the robot's behavior hides unnecessary details like initial location of the kit from the human, but important details like final location of medkit are revealed. While for the second user study, we used the illustration in Figure 3. Here the human's goal is to find *all* the medkits on the floor. In this case, the robot while picking up items necessary for its goal, also picks up the medkits in rooms A and B (recall that the robot knows the item locations). Further, it reveals to the human that rooms A to D are empty. The robot then drops one medkit in room E and takes the other medkit by itself. Thereby, hiding complexities like existence of multiple medkits, their initial locations, while revealing information that rooms A to D are empty allowing the human to deduce that the medkits (if any) would be in room E.

In hypothesis 1, our aim is to check whether the legible actions allow the robot to ensure human's awareness of the assistance. From results of questions (3) and (4) shown in Figure 4a, we can see that for the baseline behaviors, only 6 (combining both options (b) and (c) referring to assistive behavior) out of 31 participants and 6 out of 27 participants attributed assistive behavior to the robot in study 1 and 2 respectively. In contrast, for behaviors with one extra legible action, 25 out of 31 participants and 24 out of 27 participants attributed assistive behavior to the robot in study 1 and 2 respectively. Since the only difference between the two scenarios is a single legible action, the results confirm our hypothesis that legible actions make **H** aware of **R**'s assistance.

In hypothesis 2, our aim is to check whether the assistance provided by **R** allows **H** to experience potential reduction in task load and the overhead of processing robot's behavior and coming with a plan to solve the task. For first study, the average score for workload for the baseline behavior was 3.22 (recall that 1 denotes "very hard") in contrast to that of 5.96 for PA (proactive assistant), with a statistical significance (p-value = 0.0000001, p-value $< 0.05$) obtained by running a two tailed paired t-test, an effect size of 1.89 by running Cohen's d test. While the average score for processing robot's behavior was 3.45 for baseline and 6.06 for PA, with a p-value $< 0.05$ and effect size of 1.62. For second

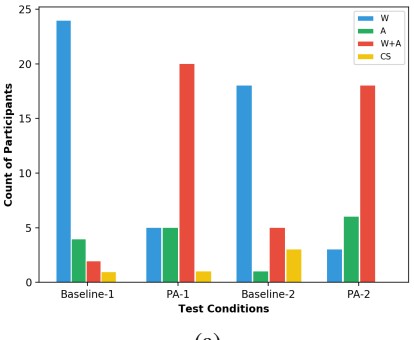

(a)

| Domain | # | $\mathcal{C}_{\mathbf{H}}(\pi^*_{\mathbf{H}})$ | $\alpha = 0.2$ | | | $\alpha = 0.8$ | | | m | Time (sec) |
|---|---|---|---|---|---|---|---|---|---|---|
| | | | $\mathcal{C}_{\mathbf{H}}(\pi_{\text{MA-COPP}})$ | % decrease | $|\pi_{\text{MA-COPP}}|$ | $\mathcal{C}_{\mathbf{H}}(\pi_{\text{MA-COPP}})$ | % decrease | $|\pi_{\text{MA-COPP}}|$ | | |
| Driverlog | 1 | 7 | 3 | 57.14 | 8 | 4 | 42.85 | 6 | 7000 | 155 |
| | 2 | 8 | 5 | 37.5 | 11 | 5 | 37.5 | 7 | 7000 | 171 |
| | 3 | 9 | 3 | 66.67 | 9 | 4 | 55.55 | 7 | 7000 | 165 |
| USAR | 1 | 15 | 3 | 80 | 14 | 5 | 66.67 | 8 | 11000 | 257 |
| | 2 | 15 | 4 | 73.33 | 12 | 7 | 53.33 | 10 | 11000 | 240 |
| | 3 | 12 | 4 | 66.67 | 13 | 4 | 66.67 | 7 | 11000 | 254 |

(b)

Figure 4: (a) Results for Hypothesis 1. The four colors stand for 4 options in questions (3) and (4). Here PA refers to Proactive Assistant, and 1 and 2 denote the user study numbers. (b) Empirical evaluation results for two domains with for different $\alpha$ values (shows human prioritizing between processing load vs task load).

study, the average score for workload was 2.74 for baseline in contrast to that of 5.55 for PA, with a p-value $< 0.05$ and effect size of 1.95. The average score for processing robot's behavior was 3.55 for baseline in contrast to 5.85 for PA, with a p-value $< 0.05$ and effect size of 1.67. All effect sizes suggest each of the two conditions differ by a large standard deviation. This confirms our hypothesis that the legible and obfuscating actions reduce the overall task and processing load by hiding unnecessary complexities and revealing necessary information.

## 6 Conclusion

We proposed a framework for settings where an AI agent can proactively assist a human. We discussed how an AI agent can reason over the human's awareness of the assistance by modulating her belief states to reveal necessary information and hide irrelevant information. Specifically, we propose a set of guidelines that allow the agent to play the role of a *proactive assistant*. We then discuss a solution approach for quickly sampling partial joint plans and constructing a utility tree to synthesize desired assistive behaviors. Through user study and empirical evaluations we validate our hypotheses and analyze the performance of our approach.

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
