# OpenReview forum: "Planning for Proactive Assistance in Environments with Partial Observability"
_icaps-conference.org/ICAPS/2021/Workshop/XAIP — XAIP 2021_

### Official Review · AnonReviewer1 · 2021-07-04
**Interesting ideas on proactive assistance**

**Rating:** 7
**Confidence:** 3

**Review:**


The paper presents an algorithm for an autonomous planning-based agent to proactively help another agent (i.e. Human) to perform their task while trying to achieve its own goals. This proactive assistant is such that does help the human agent achieve their goals in a more efficient and legible manner. The problem is presented as a multi-agent controlled observability planning problem, where the human agent observability is modelled as belief states. The proposed MCTS-based algorithm controls human's belief by either communicating information or hiding it in order to minimize the human's plan cost. An evaluation with different users shows the potential of the proposed approach.


I found the paper very interesting and of great relevance to the XAIP community. The idea of proactive assistance is novel, and the execution of this by means of explanations is a good idea. The paper is well-structured and written, but some parts could be clearer. For instance, I found the nested search part of section 3 hard to follow (i.e. what is a node in that context?). It is also not clear what is $k$ in the section "formal guidelines...". Finally, it would be interesting to see other examples or cases where the proposed proactive assistance methods could be of benefit.

While maybe not directly related to this particular research, I believe the following paper presenting the "watch-and-help" challenge may be of interest to the authors: Puig et al. "Watch-And-Help: A Challenge for Social Perception and Human-AI Collaboration", ICLR 2021.

Some minor comments/suggestions:
- I kind of missed the "introduction" section title
- In section 2.1, it is not clear how do observations $\Omega_H$ are represented.
- In section 3, what is "phase 2"?
- In section 3, the UCT bound reads a bit strange as it seems to mean node - utility (node minus utility) while I understand it means "node utility".
- The algorithm is missing some definitions: it is not clear what are $\beta$ and $\sigma$, nor what is ${b_\mathcal{T}^\pi}^H$.
- Section 4 mentions supplementary material which I was not able to find.
- Why are the hypotheses 1a and 1b instead of 1 and 2?
- In the user study section, it mentions one scenario "satisfied 3". It is not clear what is the 3 things that are satisfied.
- It could be good to add which was the filter question.
- Maybe the user study questions could be summarised in a table for easier referencing while reading the text.
- It is not clear what was the baseline behavior. I assume it was the behavior without the explanations, but this should be made explicit.
- Some grammatical issues:
 - "aware of the both" -> "aware of both"
 - sentence in section 2.1 starting with "At any time step" is strangely phrased
 - "an minimum cost" -> "a minimum cost"
 - "in first part of the joint" -> "in the first part of the joint"
 - "both the domains" -> "both domains" reads a bit better (appears multiple times)

---

### Official Review · AnonReviewer2 · 2021-07-06
**The proposed framework for proactive join planning agents shows promise but lacks the necessary commentary that links the work to explainability.**

**Rating:** 5
**Confidence:** 3

**Review:**

This paper describes a planning agent that can provide proactive task assistance to humans. When the agent and the human have partial knowledge about their tasks,  by modeling and updating beliefs. Authors proposed a formal definition of this framework (called MA-COPP) incorporating human’s planning problem, agents' and humans action models and observations. Objective of this framework is to find a joint plan that minimises the cost of the human. The evaluation was carried out in the USAR and the DriverLog domains computationally and further evaluation was done with a user study.

Is papers proposed solution is defined and formalised well. The running example given early in the paper helps in understanding the problem and the aim of the authors, but it is less helpful in grounding some of the formalisation.

Two user studies were conducted using a within-subject design. Results indicate the author's proposed proactive agent managed to reduce the task load for the human with statistically significant margins.

The major drawback of this paper is that it is only loosely tied to explainability. While the problem of joint task planning can definitely be improved using explainability methods, I found no commentary that links explainability with the framework that is proposed. I also found the novelty and the positioning of the work relative to related work less refined. As this workshop focuses mainly on explainability, I strongly suggest authors modify the narrative to comment on the relation with explainable planning.

---

### Meta-Review · Area_Chairs · 2021-07-07

**Recommendation:** Accept
**Confidence:** 5

**Metareview:**

Thank you for your submission.

Reviewers agree that the ideas presented in the paper are interesting, the solution is well defined and the user study is of value.
The reviewers also agree that this area of work (human aware proactive assistance) is of relevance to the XAI community. However, AnonReviewer2 believes that the paper should be adapted to focus more on explicit explanations and ground the work with respect to existing work in XAI. Whilst I agree with the second point, I believe that this work is currently still of great relevance to the XAI community and this workshop and certainly fits the themes of legibility and model reconciliation.

Comments:
- Although it is usually difficult to compare the results of a user study (as there aren't many out there for a given XAI topic), it is easier to compare empirical results, which would be good to see.
- It would be interesting to see this work extended with explicit explanations, especially for more complex domains where it would likely be more difficult to update the human's beliefs by acting legible (or obfuscating) alone.
- It would be interesting to have a more in-depth analysis of whether legibility or obfuscation is better at reducing human work, and under which conditions.

We hope that these reviews will be helpful in developing this work further. Please consider the comments for the camera-ready version. We look forward to your presentation!

---

### Decision · Program_Chairs · 2021-07-08

Accept